# Association of clinical factors with survival outcomes in laryngeal squamous cell carcinoma (LSCC)

Pei Yuan Fong[1], Sze Huey Tan[2], Darren Wan Teck Lim[1], Eng Huat Tan[1], Quan Sing Ng[1], Kiattisa Sommat[3], Daniel Shao Weng Tan[1], Mei Kim Ang[1] *

1 Division of Medical Oncology, National Cancer Centre, Singapore, 2 Division of Clinical Trials and Epidemiological Sciences, National Cancer Centre, Singapore, 3 Division of Radiation Oncology, National Cancer Centre, Singapore

* Ang.mei.kim@singhealth.com.sg

**Data Availability Statement:** All relevant data are within the manuscript.

**Funding:** The authors received no specific funding for this work.

## Abstract

### Aim

Treatment strategies in laryngeal squamous cell cancer (LSCC) straddle the need for long term survival and tumor control as well as preservation of laryngeal function as far as possible. We sought to identify prognostic factors affecting LSCC outcomes in our population.

### Methods

Clinical characteristics, treatments and survival outcomes of patients with LSCC were analysed. Baseline comorbidity data was collected and age-adjusted Charlson Comorbidity Index (aCCI) was calculated. Outcomes of overall survival (OS), progression-free survival (PFS) and laryngectomy-free survival (LFS) were evaluated.

### Results

Two hundred and fifteen patients were included, 170 (79%) underwent primary radiation/chemoradiation and the remainder upfront surgery with adjuvant therapy where indicated. The majority of patients were male, Chinese and current/ex-smokers. Presence of comorbidity was common with median aCCI of 3. Median OS was 5.8 years. On multivariable analyses, high aCCI and advanced nodal status were associated with inferior OS (HR 1.24 per one point increase in aCCI, P<0.001 and HR 3.52; p<0.001 respectively), inferior PFS (HR 1.14; p = 0.007 and HR 3.23; p<0.001 respectively) and poorer LFS (HR 1.19; p = 0.001 and HR 2.95; p<0.001 respectively). Higher tumor (T) stage was associated with inferior OS and LFS (HR 1.61; p = 0.02 and HR 1.91; p = 0.01 respectively).

### Conclusion

In our Asian population, the presence of comorbidities and high nodal status were associated with inferior OS, PFS and LFS whilst high T stage was associated with inferior LFS and OS.

**Competing interests:** The authors have declared that no competing interests exist.

## Introduction

More than half a million patients worldwide are diagnosed with head and neck squamous cell carcinoma (HNSCC) every year, of which one quarter occur in the larynx [1]. In USA alone, approximately 3660 patients died from laryngeal cancer in 2017 [2]. Known risk factors for laryngeal cancer are tobacco and alcohol consumption with linear association of both with the development of laryngeal cancer [3]. However, despite reduction in smoking rates leading to a reduction in incidence of laryngeal cancer, the 5-year survival rate has not changed significantly over past 40 years and still remains at about 60% [2,4].

Management of laryngeal squamous cell carcinoma (LSCC) is particularly challenging, due to the substantial functional morbidity and psychosocial effects associated with laryngectomy, and thus, the need to balance between optimal tumor control whilst preserving organ function as far as possible. The mainstay of treatment for early stage LSCC is radiation therapy or laryngeal preservation surgery, both of which give good rates of local control [5]. Up till the early 1990s, total laryngectomy had been the mainstay of therapy for locally advanced laryngeal cancer. However, the landmark Department of Veterans Affairs (VA) laryngeal study changed the treatment paradigm by evaluating the role of sequential induction chemotherapy followed by radiotherapy in comparison to standard upfront laryngectomy followed by post-operative radiotherapy in the stage III/IV LSCC. There was no significant difference in survival outcome and a laryngeal preservation rate of 64% in the non-surgical arm [6]. A second landmark study by the Radiation Therapy Oncology Group (RTOG) compared an alternate strategy of concurrent cisplatin chemotherapy with radiotherapy to induction chemotherapy followed by radiotherapy, or radiotherapy alone, in operable stage III/ IV LSCC. Both induction chemotherapy and concurrent chemoradiotherapy (CRT) significantly improved laryngectomy-free survival (LFS) compared with radiation (RT) alone [7,8]. A study performed in our center compared primary CRT with surgery followed by adjuvant RT in patients with stage III/IV HNSCC, of which about 30% of patients had LSCC [9,10]. Whilst the study closed prematurely due to poor accrual, analysis of patients recruited demonstrated no difference in disease-free survival between both treatment arms and with overall organ preservation rate of 45%. Thus, non-surgical approaches incorporating chemotherapy with radiotherapy have now been established as the standard of care for locally advanced laryngeal cancer as they enable preservation of the larynx whilst achieving meaningful long-term survival in a proportion of patients.

However, not all patients are suitable for these organ-preserving approaches. About one third of patients treated with these organ-preservation approaches do ultimately still experience relapse locally and/or distant metastatic disease. In the VA study, salvage laryngectomies were required more often in patients with gross cartilage invasion, stage IV cancers and T4 tumors [6] and of note, patients with 'large volume' T4 tumors (defined as tumors penetrating through the cartilage or extending more than 1cm in to the base of the tongue) were excluded from the RTOG study [7]. Other studies have demonstrated that patients with T4 and stage IV disease who undergo upfront total laryngectomy have better outcomes compared with CRT [11–13]. Patients with significant pulmonary compromise, have poor laryngeal or swallowing function and thus who are at a high risk for aspiration are also poor candidates for conservative treatments and may be better treated with upfront surgery.

Presence of significant comorbidities is common in HNSCC with about 30–50% of patients having at least 1 comorbidity [14–19]. Various indices such as the Charlson comorbidity index (CCI) and ACE-27 have been validated for use in the head and neck cancer population [20–23]. Studies have consistently reported that the presence of comorbidities is associated with poorer overall survival in HNSCC [15–19,24,25] and in particular in LSCC [14,26,27]. Many studies report an effect of comorbidity on OS only, and not cancer-specific survival

[14,15,17,19], postulating that this is due to higher non-cancer related mortality, which could affect about between 10–30% of patients [28–31] despite control of their cancer. However, other studies report poorer cancer-specific survival in association with comorbidity [27,32], which was attributed to various factors including its detrimental effect on surgical outcomes, higher risk of severe complications during treatment [24,27,32,33], or due to comorbidity leading to a change in therapeutic decisions in up to about 20–30% of patients [34] and/or the use of substandard therapy [35]. Significantly, CCI and Eastern Cooperative Oncology Group (ECOG) performance status are not that well correlated as ECOG score does not account for degree and severity of comorbidity [15,24]. Thus, comorbidities have an impact on survival not only by influencing treatment decisions by physicians and patients, and patients' tolerance of treatment in the shorter term, but also by contributing to risk of longer term competing non-cancer related mortalities and/or second primary tumors.

Hence, there remains a need to better characterize risk factors such as comorbidity and their effect on treatment outcomes in LSCC. In this study, we aim to describe survival outcomes in patients with LSCC in our population, and to identify prognostic factors associated with outcomes.

## Methodology

### Patient populations and treatment

Patients diagnosed with LSCC between 1994 and 2013 inclusive and treated at the National Cancer Centre Singapore were identified and their clinical records were reviewed. Patients were staged according to the AJCC cancer staging manual, 7th edition. Patients who did not complete prescribed treatment at our centre, defaulted follow-up within 1 year, or who were palliatively managed were excluded from analyses. Decisions regarding treatments were made by individual physicians based on recommendations from a multidisciplinary tumour board. No patient underwent induction chemotherapy. Patients were followed up according to our institution's standard practice. This study was approved by the Singhealth Centralised Institutional Review Board. Waiver of consent was obtained for this study as this was a retrospective analysis on anonymised data.

### Comorbidity assessment

The age-adjusted Charlson comorbidity index (aCCI) was calculated for all patients. The Charlson comorbidity index (CCI) is a weighted measure that incorporates 19 different medical categories, each weighted according to its potential to impact on mortality [20]. The aCCI is calculated by adding 1 point to the CCI score for every decade over 40 [21]. Information on pre-existing comorbidities was derived from the secondary diagnoses coded according to the international classification of diseases, 10th revision, and from the patients' charts. The index head and neck cancer was not coded as comorbidity.

### Time to event measurements

Duration of overall survival (OS) was measured from the date of diagnosis to the date of death, or the date of last follow-up for surviving patients. Patients alive at time of analysis were censored at the date of last follow up. Death data was obtained from patients' medical records as well as Singapore Death Registry. Duration of progression-free survival (PFS) was measured from the date of diagnosis to the date of progression or death (if no progression was reported before death) or the date of last follow-up. Patients alive with no progression were censored at date of last follow up. Duration of laryngectomy-free survival (LFS) was measured from the

date of diagnosis to the date of salvage laryngectomy or death (if no laryngectomy was performed) or the date of last follow-up. Patients who were alive without a laryngectomy were censored at date of last follow up.

## Statistical analysis

Baseline categorical variables were summarised as frequency and percentage, and continuous variables were summarised as median with inter-quartile range (IQR) and range. Comparisons of patient demographics and clinical characteristics by treatment (RT and CRT) were performed using Fisher's exact test or chi-squared test for categorical variables (where appropriate) and Mann-Whitney U test for continuous variables. The chi-squared test for trend was used to test if there was a linear trend relating aCCI score to treatment decision (single/combination modality).

Survival curves were estimated by Kaplan-Meier method and median survival time was reported with 95% confidence interval (95% CI). The log-rank test was used to determine if there was a difference in survival curves between different groups of patients. Univariable and multivariable analyses were performed using the Cox proportional hazards model. Patient demographics and baseline disease characteristics associated with survival in the univariable cox regression model with a significance level of $P<0.1$ were included for model selection. Variable selection was performed using a backward selection strategy using the likelihood ratio test with $P<0.05$ as the criteria for inclusion in the final multivariable model. Treatment effect of RT versus CRT (in terms of OS, PFS and LFS) were estimated using multivariable Cox regression model, adjusting for prognostic factors and baseline variables that were significantly different between the two treatments and also associated with the outcome. Proportional hazard assumption for using the Cox regression model was assessed using the Schoenfeld residuals test. A two-sided p-value less than 0.05 was considered statistically significant. All analyses were performed in STATA version 15.0.

## Results

### Patient cohort demographics

Two hundred and fifteen patients diagnosed with LSCC and treated with curative intent between 1994 and 2013 were included. The majority of patients were male (94.4%), Chinese (78.6%), and current or ex-smokers (80%), with a median age of 67 years (range 36–94). At diagnosis, the most common symptoms were a hoarse voice (93.5%), followed by pain (19.9%) and dysphagia (16.2%). Most patients had a good performance score of ECOG 0 (60.9%) or ECOG 1 (36.3%), with a median aCCI of 3 (range, 0–12). The characteristics of patients are shown in Table 1.

### Treatment

One hundred and seventy patients (79%) underwent primary RT or CRT, while 45 (21%) patients were treated by surgery followed by adjuvant RT/CRT where indicated.

**Chemotherapy.**   The majority of patients undergoing chemotherapy received cisplatin-based chemotherapy (n = 42/49; 85.7%), administered 3-weekly during radiotherapy for 3 cycles at $100mg/m^2$ unless contra-indicated due to inadequate renal function, in which case they received paclitaxel, or carboplatin, or cetuximab instead. The mean cumulative dose of cisplatin administered was $255mg/m^2$.

**Radiotherapy.**   All patients were immobilised with a customised thermoplastic mask and treated with 6MV photon encompassing the primary bearing area and regional lymph nodes.

**Table 1. Clinical characteristics of patients.**

|  | Frequency (%) |
|---|---|
|  | **N = 215** |
| **Age at diagnosis, years** |  |
| Mean (SD) | 67 (10.8) |
| Median (IQR) | 67 (59.5, 73.9) |
| Range | 36 to 94 |
| **Age at diagnosis** |  |
| <65 years old | 95 (44.2) |
| ≥65 years old | 120 (55.8) |
| **Gender** |  |
| Male | 203 (94.4) |
| Female | 12 (5.6) |
| **Race** |  |
| Chinese | 169 (78.6) |
| Malay | 18 (8.4) |
| Indian | 18 (8.4) |
| Others | 10 (4.7) |
| **Smoking status** |  |
| Non-smoker | 39 (18.1) |
| Smoker | 121 (56.3) |
| Ex-smoker | 51 (23.7) |
| Unknown | 4 (1.9) |
| **Stage at diagnosis** |  |
| Stage 1 | 63 (29.3) |
| Stage 2 | 43 (20.0) |
| Stage 3 | 38 (17.7) |
| Stage 4a | 68 (31.6) |
| Stage 4b | 3 (1.4) |
| **T-stage at diagnosis** |  |
| T1 | 68 (31.6) |
| T2 | 59 (27.4) |
| T3 | 36 (16.7) |
| T4 | 52 (24.2) |
| **N-stage at diagnosis** |  |
| N0 | 155 (72.1) |
| N1 | 23 (10.7) |
| N2 | 35 (16.3) |
| N3 | 2 (0.9) |
| **ECOG** |  |
| 0 | 131 (60.9) |
| 1 | 78 (36.3) |
| 2 | 5 (2.3) |
| 3 | 1 (0.5) |
| **aCCI score** |  |
| Median (IQR) | 3 (2, 4) |
| Range | 0 to 12 |
| **aCCI score** |  |
| 0–3 | 143 (66.5) |

(*Continued*)

**Table 1.** (Continued)

| | Frequency (%) |
|---|---|
| | N = 215 |
| 4–12 | 72 (33.5) |

SD: standard deviation

IQR: interquartile range

ECOG: Eastern Cooperative Oncology group performance status

aCCI: age-adjusted Charlson Comorbidity Index

Patients who received upfront radical RT were treated with 66–70 Gy delivered in 33–35 fractions, whereas patients who received adjuvant RT were treated with 60–66 Gy in 30–33 fractions. Patients with T1N0 and some patients with T2N0 received 55Gy in 20 fractions.

The gross tumour volume (GTV) was defined as any visible gross disease based on radiological, endoscopic and clinical findings. High risk clinical target volume (CTV) is an expansion of 5–10 mm margin around the GTV, and with editing off natural tumour barriers. This volume was treated with 66–70 Gy. Intermediate risk CTV included the possible local subclinical infiltration of the primary site as well as first echelon nodal stations and was prescribed 60 Gy. Low risk CTV included regional nodal stations which are not first echelon nodes and were not adjacent to the levels of involved nodes and was prescribed 50 Gy. In the adjuvant setting, the tumour bed and involved nodal stations were treated to 60 Gy with a further 6 Gy boost to areas of extracapsular extension or close/positive margin.

Most patients before 2005 were treated with 2-dimensional (2D) or 3-dimensional conformal technique whereas the latter patients were treated with intensity modulated radiotherapy (IMRT). In the IMRT technique, all dose levels were delivered within the same plan with the higher doses effected through a simultaneous integrated boost. In the 2D technique, the RT was delivered via 2 shaped lateral parallel opposed fields with a low anterior neck match in 2 or 3 phases using shrinking field technique. Compliance to radiotherapy was high overall, with almost all patients (197/201, 98%) completing RT.

**Treatment modality according to stage and aCCI.** The majority of patients who underwent upfront surgery had T4 disease (71.1%, 32 out of 45 patients), however there were no other demographic differences amongst the patients undergoing upfront surgery vs non-surgical treatment. Of 52 patients with T4 disease, 32 (61.5%) had upfront surgery whilst the remainder had RT or CRT (38.5%).

Almost all of the patients with stage 1/2 disease received single modality treatment with primary RT alone (101/106; 95.3%), of the remainder, 4 patients (3.8%) received surgery alone and 1 received surgery followed by RT. Amongst 109 stage 3/4 patients, two-thirds (73 patients, 67%) received combined modality treatment: in particular, 43 (39.4%) received primary CRT, 24 (22.0%) underwent surgery followed by adjuvant RT whilst 6 patients (5.5%) had surgery followed by adjuvant CRT. The remaining one-third of patients (n = 36, 33%) received single modality treatment (surgery or RT): 26 (23.9%) received radiation alone: of these, 12 (46%) were elderly (>70 years old), 9 (35%) had multiple comorbidities; these patients were deemed unsuitable for CRT, whilst 4 (15%) declined chemotherapy and the remaining patient was treated in 1997, when CRT was not yet standard of care. 10 patients underwent surgery alone: 3 declined adjuvant RT, 3 had received previous RT to that region for a separate unrelated tumour (nasopharyngeal carcinoma), 2 patients defaulted RT, 1 patient had a synchronous carcinoma of the colon requiring treatment, and 1 had post-operative wound infection which precluded RT.

**Table 2. Association of aCCI score with treatment in stage 3/4 patients.**

| | | | | | aCCI score | | | |
|---|---|---|---|---|---|---|---|---|
| | Total | 0 | 1 | 2 | 3 | 4 | 5–12 | p-value |
| Treatment | | | | | | | | <0.001 (<0.001#) |
| Single modality (RT or surgery alone) | 36 (33.0) | 0 (0.0) | 3 (16.7) | 3 (11.5) | 7 (31.8) | 11 (47.8) | 12 (80.0) | |
| Combination modality (Surgery->Adj-RT included) | 73 (67.0) | 5 (100.0) | 15 (83.3) | 23 (88.5) | 15 (68.2) | 12 (52.2) | 3 (20.0) | |

P-value calculated using Fisher's exact test

#P-value for test of trend calculated using chi-squared test for trend

Amongst stage 3/4 patients, the treatment modality (single vs combined) was associated with the aCCI score at diagnosis (p<0.001) and a test for trend showed evidence of a linear trend, with the proportion of patients receiving single modality treatment increasing as the aCCI score increased (p<0.001) (Table 2).

## Survival Outcomes

**Overall survival (OS).** With a median follow up of 4.1 years (Range: 0.14 to 14.84 years), one hundred out of 215 patients (46.5%) were alive, and the median OS time was 5.8 years (95% CI: 4.83 to 7.14 years). Kaplan-Meier curves for OS for all patients, by N-stage and by age-adjusted CCI are presented in Fig 1A, 1B and 1C respectively.

In univariable analysis, age at diagnosis, clinical stage, T-stage, N-stage, ECOG and aCCI score were associated with OS. In multivariable analysis, aCCI score, gender, T-stage and N-stage remained significantly associated with OS (Table 3). Patients with N2/N3 disease were at higher risk of death (HR = 3.52, 95% CI: 2.23 to 5.55; p<0.001) than patients with N0/N1 disease. Increasing aCCI score was also significantly related to OS (HR 1.24 per 1-point increase; 95% CI 1.12–1.38; p<0.001).

**Progression-free survival (PFS).** With a median follow up of 4.10 years, 138 patients (64%) experienced disease progression or died. The median PFS was 3.2 years (95% CI: 2.44 to 4.52 years). PFS curves for all patients, by N-stage and by age-adjusted CCI are presented in Fig 2A, 2B and 2C respectively.

In univariable analysis, clinical stage, N-stage and aCCI score were associated with PFS. However, in the multivariable analysis, only aCCI score and N-stage remained significantly associated with PFS (Table 4). Patients with N2/N3 disease or high aCCI score were at higher risk of experiencing progressive disease or death (HR = 3.23, 95% CI: 2.13 to 4.90; p<0.001 and HR 1.14; 95% CI: 1.04–1.25; p = 0.007 respectively).

## Organ preservation for laryngeal cancer

**Patient characteristics.** One hundred and seventy out of 215 patients (79.0%) underwent first-line RT or CRT. Compared with patients treated with RT alone, patients treated with CRT were on average younger (age <65 years, 67.4% vs 37% respectively, p<0.001), had lower aCCI score (aCCI ≤3, 86% vs 59.1%, p = 0.001) and had higher disease stage (stage 3/4, 100% vs 20.5%); T3/4, 69.8% vs 15.7% and N2/3, 41.8% vs 3.9%, p<0.001.

**Laryngectomy-free survival (LFS).** One hundred and nine out of 170 patients (64%) had salvage laryngectomy or had died at the time of analysis. The median LFS was 3.7 years (95% CI: 2.63 to 5.17 years). Kaplan-meier curves for LFS for all patients undergoing RT/ CRT by N-stage and by age-adjusted CCI are presented in Fig 3A, 3B and 3C respectively.

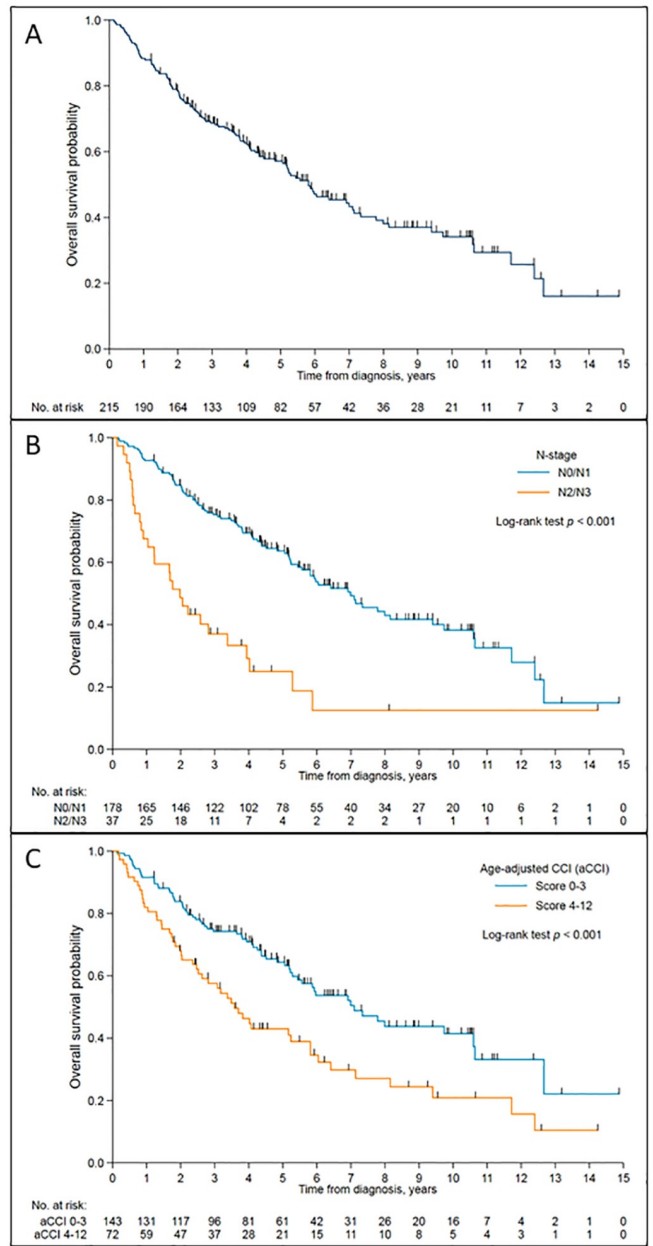

**Fig 1. Kaplan-Meier survival curve of overall survival probability against time.** (A) in the entire cohort. (B) according to nodal stage. (C) according to age-adjusted Charlson comorbidity index (aCCI).

In univariable analysis, overall stage, T-stage, N-stage, and aCCI score were associated with LFS. In the multivariable analysis, aCCI score, T-stage and N-stage remained significantly associated with LFS (Table 5). Patients with T4 disease or N2/N3 were at higher risk of losing their laryngeal function due to a laryngectomy or death (HR 1.91, 95% CI 1.14–3.18; p = 0.01 and HR = 2.95, 95% CI: 1.75 to 4.98; p<0.001 respectively) than patients with T1-3 and/or N0/N1 disease.

**Table 3. Univariable and multivariable overall survival results for patient characteristics.**

| | No. of events / patients | Univariable | | Multivariable | |
|---|---|---|---|---|---|
| | | OS Hazard ratio (95% CI) | p-value | Adjusted Hazard ratio (95% CI) | p-value |
| | 116/215 | | | | |
| Age at diagnosis, per 10 years | 116 / 215 | 1.40 (1.16 to 1.69) | <0.001 | | |
| Gender | | | | | |
| Male | 112 / 203 | 1 | | | |
| Female | 4 / 12 | 0.41 (0.15 to 1.11) | 0.08 | 0.25 (0.09 to 0.70) | 0.008 |
| Smoking status | | | | | |
| Non-smoker | 17 / 39 | 1 | | | |
| Smoker | 67 / 121 | 1.07 (0.63 to 1.83) | 0.8 | | |
| Ex-smoker | 30 / 51 | 1.35 (0.74 to 2.44) | 0.3 | | |
| Unknown | 2 / 4 | 2.20 (0.51 to 9.57) | 0.3 | | |
| Stage | | | | | |
| Stage 1 | 28 / 63 | 1 | | | |
| Stage 2 | 25 / 43 | 1.47 (0.85 to 2.53) | 0.2 | | |
| Stage 3 | 17 / 38 | 1.20 (0.66 to 2.20) | 0.5 | | |
| Stage 4a/4b | 46 / 71 | 2.22 (1.38 to 3.56) | 0.001 | | |
| T-stage | | | | | |
| T1-T3 | 82/163 | 1 | | | |
| T4 | 34 / 52 | 1.74 (1.16 to 2.60) | 0.007 | 1.61 (1.07 to 2.43) | 0.02 |
| N-stage | | | | | |
| N0/N1 | 88/178 | 1 | | | |
| N2/N3 | 28/37 | 3.03 (1.96 to 4.67)[#] | <0.001 | 3.52 (2.23 to 5.55) | <0.001 |
| ECOG | | | | | |
| 0 | 55 / 131 | 1 | | | |
| 1–3 | 61 / 84 | 1.57 (1.08 to 2.28) | 0.02 | | |
| aCCI score per 1 score increase | 116 / 215 | 1.19 (1.08 to 1.31) | <0.001 | 1.24 (1.12 to 1.38) | <0.001 |
| aCCI score | | | | | |
| 0–3 | 66 / 143 | 1 | | | |
| 4–12 | 50 / 72 | 1.86 (1.29 to 2.69) | 0.001 | | |

P-value calculated using Wald test (from Cox model)

[#] Proportional hazard assumption violated

OS: overall survival

CI: Confidence interval

ECOG: Eastern Cooperative Oncology Group

aCCI: age-adjusted Charlson Comorbidity Index

## Patterns of relapse after primary RT/CRT

Seventy-three patients relapsed after primary RT/CRT, the majority (66 patients, 90.4%) had local relapse in the larynx and/or lymph nodes, of which 50 patients (71%) underwent curative treatments. 16 patients with local relapse did not undergo curative therapy either due to unsuitability or patient's choice. Seven patients relapsed with distant metastases (4 with distant metastases and 3 with both local and distant relapse).

## Treatment effect of CRT and RT

After accounting for significant prognostic factors, there was a significant treatment effect favoring CRT over RT in terms of OS, PFS and LFS (RT vs CRT, OS HR 1.96 (95% CI: 1.03–

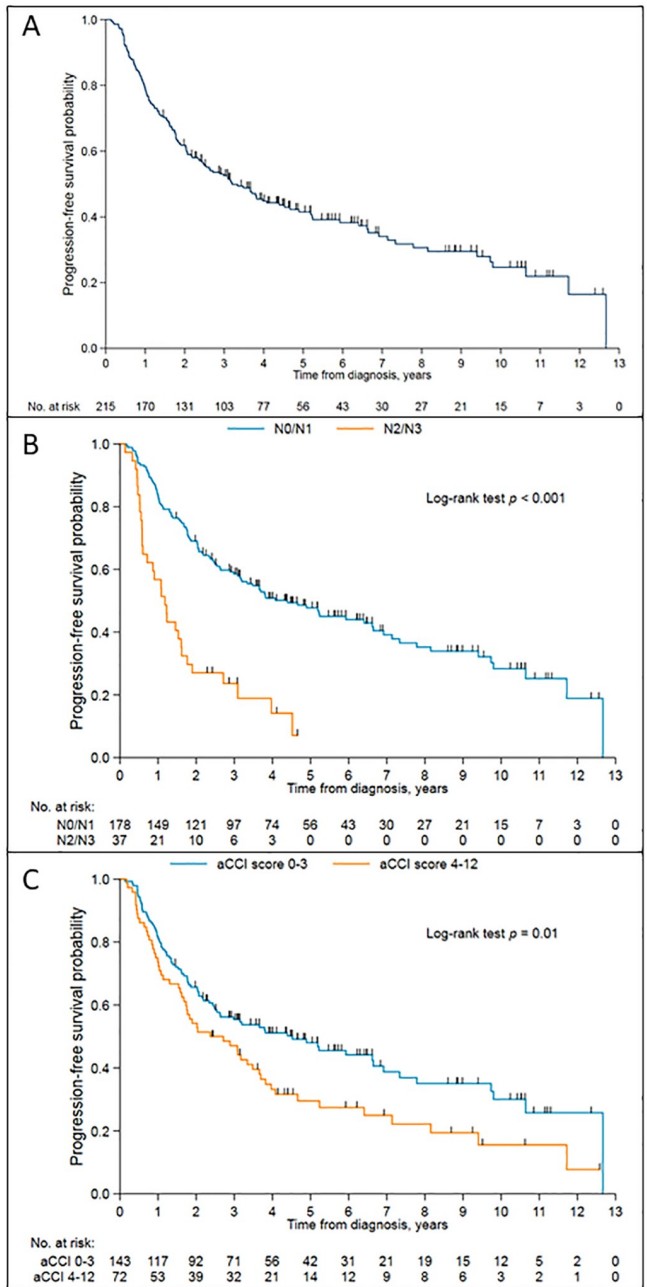

**Fig 2. Kaplan-Meier survival curve of progression free survival probability against time.** (A) in the entire cohort. (B) according to nodal stage. (C) according to age-adjusted Charlson comorbidity index (aCCI).

3.75; p = 0.04); PFS HR 2.33 (95% CI: 1.28 to 4.23; p = 0.005); LFS HR 2.14 (95% CI: 1.17 to 3.93; p = 0.01).

## Discussion

Our study demonstrates that nodal status and comorbidity status are significant factors determining PFS, OS and LFS in laryngeal cancer in our patient population. In addition, tumor stage was related to OS as well as LFS.

**Table 4. Univariable and multivariable progression-free survival results for patient characteristics.**

| | No. of events / patients | univariable | | multivariable | |
|---|---|---|---|---|---|
| | | PFS Hazard ratio (95% CI) | p-value | Adjusted HR (95% CI) | p-value |
| | 138/215 | | | | |
| Age at diagnosis, per 10 years | 138 / 215 | 1.15 (0.97 to 1.35)[#] | 0.1 | | |
| Gender | | | | | |
| Male | 132 / 203 | 1 | | | |
| Female | 6 / 12 | 0.57 (0.25 to 1.30) | 0.2 | | |
| Smoking status | | | | | |
| Non-smoker | 20 / 39 | 1 | | | |
| Smoker | 80 / 121 | 1.18 (0.72 to 1.92) | 0.5 | | |
| Ex-smoker | 35 / 51 | 1.46 (0.84 to 2.53) | 0.2 | | |
| Unknown | 3 / 4 | 3.49 (1.03 to 11.79) | 0.04 | | |
| Stage | | | | | |
| Stage 1 | 34 / 63 | 1 | | | |
| Stage 2 | 31 / 43 | 1.60 (0.98 to 2.62) | 0.06 | | |
| Stage 3 | 22 / 38 | 1.36 (0.79 to 2.33) | 0.3 | | |
| Stage 4a/4b | 51 / 71 | 1.94 (1.25 to 3.01) | 0.003 | | |
| T-stage | | | | | |
| T1-3 | 102/163 | 1 | | | |
| T4 | 36/52 | 1.26 (0.86–1.84) | 0.2 | | |
| N-stage | | | | | |
| N0/N1 | 17/178 | 1 | | | |
| N2/N3 | 31/37 | 3.15 (2.07 to 4.77) | <0.001 | 3.23 (2.13 to 4.90) | <0.001 |
| ECOG | | | | | |
| 0 | 72 / 131 | 1 | | | |
| 1–3 | 66 / 84 | 1.30 (0.93 to 1.83) | 0.1 | | |
| aCCI score per 1 score increase | 138 / 215 | 1.12 (1.03 to 1.23) | 0.01 | 1.14 (1.04 to 1.25) | 0.007 |
| aCCI score | | | | | |
| 0–3 | 83 / 143 | 1 | | | |
| 4–12 | 55 / 72 | 1.54 (1.09 to 2.17) | 0.01 | | |

P-value calculated using Wald test (from Cox model)

[#] Proportional hazard assumption violated

OS: overall survival

CI: Confidence interval

ECOG: Eastern Cooperative Oncology Group

aCCI: age-adjusted Charlson Comorbidity Index

Several previous studies have reported that nodal status is a significant prognostic factor in laryngeal cancer, with higher nodal status being associated to poorer OS [11,14,36–38] although there are limited reports of the association of nodal status and LFS. In our study, the majority of our patients were treated with non-surgical approaches, hence it is not known whether other approaches eg, upfront surgery followed by adjuvant CRT or induction chemotherapy followed by CRT will lead to better outcomes for patients with N2/N3 disease, or whether other methods of intensifying therapy will improve outcomes. However, other studies have reported that patients with high nodal status perform poorly regardless of treatment modality [6,11,37,39] and thus, these patients represent a significant challenge in terms of treatment.

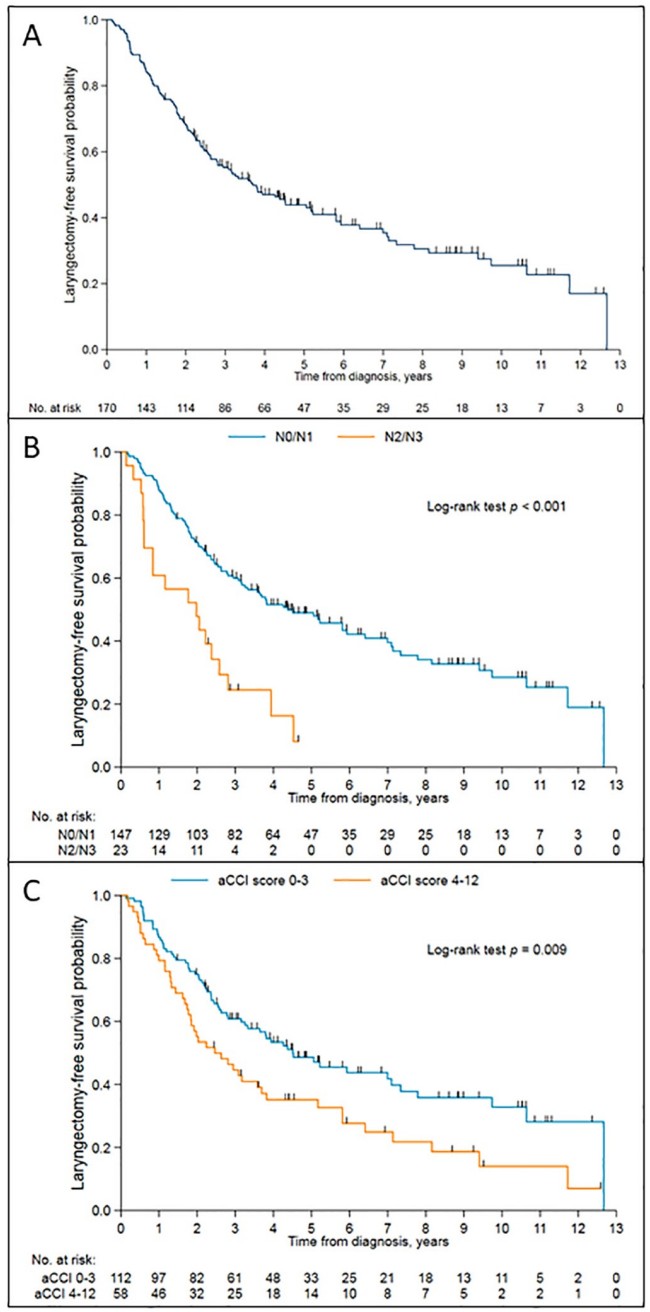

**Fig 3. Kaplan-Meier survival curve of laryngectomy free survival probability against time.** (A) in the cohort undergoing RT/CRT. (B) in the cohort undergoing RT/CRT, according to nodal stage. (C) in the cohort undergoing RT/CRT, according to age-adjusted Charlson comorbidity index (aCCI).

Our study findings of lower LFS and OS amongst patients with higher T stage are consistent with other previous reports [11,13,39] and reflect the reality that not all patients with local relapse after organ-preservation approaches such as RT/ CRT will be suitable for salvage procedures; some may have disease that is no longer resectable, or may be deemed unfit/ or decline surgery at that point. A stronger emphasis on upfront surgery in these patients with T4 disease may help improve overall survival. We were not able to compare the use of primary

**Table 5. Univariable and multivariable results for laryngectomy-free survival for patient characteristics.**

| | No. of events / patients | Univariable | p-value | Multivariable | p-value |
|---|---|---|---|---|---|
| | | LFS Hazard ratio (95% CI) | | LFS Hazard ratio (95% CI) | |
| | 109/170 | | | | |
| Age at diagnosis, per 10 years | 109 / 170 | 1.18 (0.98 to 1.41)# | 0.08 | | |
| Gender | | | | | |
| Male | 103 / 160 | 1 | | | |
| Female | 6 / 10 | 0.74 (0.33 to 1.70) | 0.5 | | |
| Smoking status | | | | | |
| Non-smoker | 17 / 33 | 1 | | | |
| Smoker | 62 / 95 | 1.17 (0.68 to 2.00) | 0.6 | | |
| Ex-smoker | 27 / 39 | 1.48 (0.81 to 2.72) | 0.2 | | |
| Unknown | 3 / 3 | 52.65 (11.99 to 231.25) | <0.001 | | |
| Stage | | | | | |
| Stage 1 | 31 / 60 | 1 | | | |
| Stage 2 | 29 / 41 | 1.64 (0.98 to 2.73) | 0.06 | | |
| Stage 3 | 19 / 33 | 1.50 (0.85 to 2.68) | 0.2 | | |
| Stage 4a/4b | 30 / 36 | 2.69 (1.62 to 4.47) | <0.001 | | |
| T-stage | | | | | |
| T1-3 | 91/150 | 1 | | | |
| T4 | 18/20 | 1.95 (1.17 to 3.24) | 0.01 | 1.91 (1.14 to 3.18) | 0.01 |
| N-stage | | | | | |
| N0-1 | 90/147 | 1 | | | |
| N2-N3 | 19/23 | 2.75 (1.65 to 4.59) | <0.001 | 2.95 (1.75 to 4.98) | <0.001 |
| ECOG | | | | | |
| 0 | 58 / 105 | 1 | | | |
| 1–3 | 51 / 65 | 1.34 (0.91 to 1.97) | 0.1 | | |
| aCCI score per 1 score increase | 109 / 170 | 1.15 (1.05 to 1.26) | 0.004 | 1.19 (1.07 to 1.31) | 0.001 |
| aCCI score | | | | | |
| 0–3 | 64 / 112 | 1 | | | |
| 4–12 | 45 / 58 | 1.66 (1.13 to 2.44) | 0.01 | | |

p-value calculated using Wald test (from Cox model)

# Proportional hazard assumption violated

LFS: Laryngectomy-free survival

CI: Confidence interval

ECOG: Eastern Cooperative Oncology group performance status

aCCI: age-adjusted Charlson Comorbidity Index

surgery vs primary RT/CRT in patients with T4 tumors within our study due to small patient numbers.

Presence of comorbidities was common in our population, this finding is similar to other studies [14–19]. In addition, aCCI was a significant prognostic factor for PFS, OS and LFS in our study. The association of inferior OS with presence of comorbidities is consistent with other studies in HNSCC [14–17,19,24–26], although there are limited data in laryngeal cancer patients regarding the impact of comorbidity on PFS and LFS. A limitation of our study is that we were not able to obtain data regarding the cause of death in the majority of patients and thus could not determine the effect of comorbidities on non-cancer and cancer-specific survival.

While RT is the mainstay of organ preserving treatment for LSCC, CRT improves laryngeal preservation rates and is the standard of care for locally advanced resectable laryngeal cancer; whilst patients with T4 disease would benefit more from an upfront surgical approach followed by adjuvant RT, with concurrent chemotherapy added in the presence of high risk features [40–42]. In our study however, one-third of stage 3/4 patients received single modality treatment due to various reasons as outlined in our results; furthermore, we have also demonstrated a linear trend of single modality treatment with increasing aCCI. Finally, our results also demonstrated that treatment of patients with RT instead of CRT was indeed associated with poorer OS, PFS and LFS. It is thus possible that the association of aCCI with inferior OS, PFS and LFS could be contributed by sub-optimal treatment in stage 3/4 patients with higher aCCI compared with patients who had lower aCCI, this has also been described in other studies [35,43].

Whilst we acknowledge this may have influenced our study findings, we were not able to directly analyze the effect of comorbidity on treatment decisions. In addition, the accuracy of comorbidity assessment depended on quality of disease coding in patient's charts and could be underestimated by missing data. Furthermore, the act of assigning patients a CCI score has the potential for error in and of itself. Although the CCI scoring system has specific rules and guidelines, there is some level of subjectivity involved in assigning CCI scores to individual patients based on a retrospective review of a patient's chart, including errors and inconsistencies in medical record keeping. This could be improved in future by standardized patient-reported comorbidity questionnaires filled out at diagnosis which have shown good correlation with standard CCI [44]. The retrospective nature of our study also means that the cohort was heterogeneous and this may further impact on treatment outcomes and study conclusions.

Management of laryngeal cancer has evolved over the past 20 years due in part to several landmark studies of chemotherapy in combination with RT [6–8,40,41]. In particular, concurrent chemoradiotherapy for locally advanced resectable laryngeal cancer only became the standard of care for organ preservation in 2003, prior to this, RT alone or sequential chemotherapy followed by RT may have been used (6). Recommendation of post-operative adjuvant concurrent CRT instead of adjuvant RT for high risk (eg, positive margins and/or extra-capsular spread) patients was established in 2004 based on the EORTC 22931 and RTOG 9501 study showing that addition of concurrent chemotherapy to adjuvant RT improved loco-regional control, progression-free survival as well as overall survival [40–42]. As our study cohort included patients treated over a 20-year period, their treatments may have been influenced by these changes in standard clinical practice, which may in turn have affected patient outcomes such as PFS, OS and LFS [6](6). Due to small patient numbers, we were not able to analyse patients' outcomes according to time period that they received treatment. Radiotherapy techniques have also evolved over time, with the introduction of IMRT in 2005 replacing 2D/3D conformal techniques, though this may not have directly affected treatment outcomes.

In summary, our study demonstrates that in our local Asian population, nodal status, T-stage, aCCI are important factors which determine treatment outcome in non-metastatic LSCC. Presence of comorbidities is common and, given its significant impact on treatment modality and survival outcomes, it is thus important to take comorbidity data into account in multi-disciplinary treatment decisions and consider routine use of a validated comorbidity scoring system in the context of stratifying patients for clinical trials. Comorbidity data may also need to be considered in the design of surveillance programs for patients after completion of their therapy. Many studies have demonstrated that a significant number of long term HNSCC survivors die from cancers other than HNSCC, and from non-cancer causes [30]. Thus whilst reducing cancer-related death remains a key priority for patients with LSCC, non-

cancer related death is also a significant hazard particularly in patients with high CCI [28–30] and routine follow-up care for HNSCC survivors should be reviewed to expand beyond surveillance for recurrent/ new HN cancers to also address this populations' specific risks, which may further optimize their OS.

As our data shows, a substantial proportion of patients with higher aCCI may be deemed unsuitable for combination therapy and/or intensive therapy, and hence, the development of better strategies to pro-actively manage the toxicities these patients may encounter, and/or the use of less intensive combined modality regimens may be helpful in improving outcomes. Patients with high nodal stage on the other hand may benefit from more intensive approaches in view of high risk of relapse and death regardless of their t-stage. By designing clinical trials targeting, or stratifying for these poor-risk subsets of patients, we may be better able to improve outcomes in LSCC. Conversely, it will also be important to identify patients with low-risk disease or good prognosis, who may benefit from de-intensified and/or organ preservation approaches, such as radiation alone, or with limited or partial laryngectomies in order to maintain optimal laryngeal function.

## Author Contributions

**Conceptualization:** Pei Yuan Fong, Eng Huat Tan, Quan Sing Ng, Daniel Shao Weng Tan, Mei Kim Ang.

**Data curation:** Pei Yuan Fong, Sze Huey Tan, Darren Wan Teck Lim, Kiattisa Sommat, Mei Kim Ang.

**Formal analysis:** Sze Huey Tan, Mei Kim Ang.

**Investigation:** Sze Huey Tan, Quan Sing Ng.

**Methodology:** Pei Yuan Fong, Sze Huey Tan, Darren Wan Teck Lim, Eng Huat Tan, Quan Sing Ng, Kiattisa Sommat, Mei Kim Ang.

**Project administration:** Mei Kim Ang.

**Supervision:** Eng Huat Tan, Daniel Shao Weng Tan, Mei Kim Ang.

**Writing – original draft:** Pei Yuan Fong, Sze Huey Tan, Mei Kim Ang.

**Writing – review & editing:** Pei Yuan Fong, Sze Huey Tan, Darren Wan Teck Lim, Eng Huat Tan, Quan Sing Ng, Kiattisa Sommat, Daniel Shao Weng Tan, Mei Kim Ang.

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
