## [Decision Letter · Decision Letter 0]

19 Aug 2019

PONE-D-19-20376

Association of clinical factors with survival outcomes in laryngeal squamous cell carcinoma

PLOS ONE

Dear Dr. Mei-Kim Ang,

Thank you for submitting your manuscript to PLOS ONE. After careful consideration, we feel that it has merit but does not fully meet PLOS ONE’s publication criteria as it currently stands. Therefore, we invite you to submit a revised version of the manuscript that addresses the points raised during the review process.

ACADEMIC EDITOR: Please kindly respond to the reviewers' comments, especially the various treatment strategies. I also suggest having some discussion about the changes in cancer care within the 20 years.   

We would appreciate receiving your revised manuscript by Oct 03 2019 11:59PM. To enhance the reproducibility of your results, we recommend that if applicable you deposit your laboratory protocols in protocols.io, where a protocol can be assigned its own identifier (DOI) such that it can be cited independently in the future. For instructions see: http://journals.plos.org/plosone/s/submission-guidelines#loc-laboratory-protocols

We look forward to receiving your revised manuscript.

Kind regards,

Jason Chia-Hsun Hsieh, M.D. Ph.D

Academic Editor

PLOS ONE

Journal Requirements:

2. Please amend either the title on the online submission form (via Edit Submission) or the title in the manuscript so that they are identical.

Additional Editor Comments:

Please kindly respond to the reviewers' comments, especially the various treatment strategies. I also suggest having some discussion about the changes in cancer care within the 20 years.

Reviewers' comments:

Reviewer's Responses to Questions

5. Review Comments to the Author

Reviewer #1: This single center retrospective study started from an excellent review of the treatment for laryngeal cancer. It highlighted the unmet needs for the treatment of laryngeal cancer. The retrospective analysis also incorporates several clinical factors into the analysis of laryngeal cancer therapy and prognosis. The study is unique because of: 1) a retrospective study focusing in Asian cohort in a world-class cancer center; 2) large sample size; 3) long-term follow-up.

Several clinical factors may be considered to clarify in detail. First, what is the RT dose and technique in the cohort? What is the accumulated cisplatin dose and cisplatin schedule with RT? What is the compliance of RT in the cohort? These treatment-related factors may also have a higher impact on survival. The author may consider to incorporate it into the analysis.

Some studies showed that body weight or BMI may also be a prognostic factor for cancer therapy. Is it possible for the study to retrieve the data from medical record?

The study is worth for publication after updating these clinical data in the final results.

Reviewer #2: Retrospective and descriptive study

Easy to read paper

Too many tables

Treatment should be described according to the TNM, Stage and aCCI as there is 60% T1/T2 and 70% N0.

Already known data

Reviewer #3: In this study, Fong et al. retrospectively enrolled 215 LSCC patients who either underwent primary radiation/ chemoradiation or upfront surgery with adjuvant therapy. Comorbidities and nodal status were identified as prognostic markers. Several comments are listed as follows:

1. The major concern of this study is less novelty, since comorbidities and nodal status have both been correlated with treatment outcome in many cancers including HNSCC.

2. The treatment strategy is heterogenous and therefore make each sub-group with small number. Authors could conduct an analysis with a more specific population, e.g., locally advanced LSCC after primary chemoradiation with IMRT and triweekly cisplatin.

3. The enrollment duration is across almost 20 years, while there may be some prognosis shift either due to treatment or even general care improvement.

4. More than 2/3 of patients only accepted RT alone. As the authors discussed, they also need to explain why these patients have more comorbidities, poor performance status, or advanced age.

6. PLOS authors have the option to publish the peer review history of their article (what does this mean?). If published, this will include your full peer review and any attached files.

Reviewer #1: Yes: Hsiang-Fong Kao

---

## [Author Response · Author response to Decision Letter 0]

11 Oct 2019

Please refer to our "Response to reviewers" file if possible

Jason Chia-Hsun Hsieh, M.D. Ph.D

Academic Editor

PLOS ONE

Dear Dr. Hsieh,

Response to reviewers’ comments [PONE-D-19-20376] - [EMID:652fb23106c7bb55]

Thank you for reviewing our manuscript and for the helpful comments provided. 

Please find enclosed our response to the comments raised. The page and line numbers provided here are based on our revised manuscript with tracked changes. 

1. ACADEMIC EDITOR: Please kindly respond to the reviewers' comments, especially the various treatment strategies. I also suggest having some discussion about the changes in cancer care within the last 20 years.

 We have included a paragraph in the Discussion section (page 28) summarising the changes in management of laryngeal cancer in the past 20 years. 

 (page 28, line 497) “Management of laryngeal cancer has evolved over the past 20 years due in part to several landmark studies of chemotherapy in combination with RT (6–8,40,41). In particular, concurrent chemoradiotherapy for locally advanced resectable laryngeal cancer only became the standard of care in 2003, prior to this, RT alone or sequential chemotherapy followed by RT may have been used. Recommendation of post-operative adjuvant concurrent CRT instead of adjuvant RT for high risk (eg, positive margins and/or extra-capsular spread) patients was established in 2004 based on the EORTC 22931 and RTOG 9501 study showing that addition of concurrent chemotherapy to adjuvant RT improved loco-regional control, progression-free survival as well as overall survival (40–42). As our study cohort included patients treated over a 20-year period, their treatments may have been influenced by these changes in standard clinical practice, which may in turn have affected patient outcomes such as PFS, OS and LFS (6). Due to small patient numbers, we were not able to analyse patients’ outcomes according to time period that they received treatment. Radiotherapy techniques have also evolved over time, with the introduction of IMRT in 2005 replacing 2D/3D conformal techniques, though this may not have directly affected treatment outcomes.”

2. Reviewer #1: Several clinical factors may be considered to clarify in detail. 

(i) First, what is the RT dose and technique in the cohort? 

 Under the Results section and treatment sub-section (page 12), we have now inserted three paragraphs summarising the RT dose and technique for our study cohort: 

 (page 12, line 226) “All patients were immobilised with a customised thermoplastic mask and treated with 6MV photon encompassing the primary bearing area and regional lymph nodes. Patients who received upfront radical RT were treated with 66-70 Gy delivered in 33-35 fractions, whereas patients who received adjuvant RT were treated with 60-66 Gy in 30-33 fractions. Patients with T1N0 and some patients with T2N0 received 55Gy in 20 fractions. 

The gross tumour volume (GTV) was defined as any visible gross disease based on radiological, endoscopic and clinical findings. High risk clinical target volume (CTV) is an expansion of 5-10 mm margin around the GTV, and with editing off natural tumour barriers. This volume was treated with 66-70 Gy. Intermediate risk CTV included the possible local subclinical infiltration of the primary site as well as first echelon nodal stations and was prescribed 60 Gy. Low risk CTV included regional nodal stations which are not first echelon nodes and were not adjacent to the levels of involved nodes and was prescribed 50 Gy. In the adjuvant setting, the tumour bed and involved nodal stations were treated to 60 Gy with a further 6 Gy boost to areas of extracapsular extension or close/positive margin. 

Most patients before 2005 were treated with 2-dimensional (2D) or 3-dimensional conformal technique whereas the latter patients were treated with intensity modulated radiotherapy (IMRT). In the IMRT technique, all dose levels were delivered within the same plan with the higher doses effected through a simultaneous integrated boost. In the 2D technique, the RT was delivered via 2 shaped lateral parallel opposed fields with a low anterior neck match in 2 or 3 phases using shrinking field technique.”

(ii) What is the accumulated cisplatin dose and cisplatin schedule with RT? 

 Under the Results section and treatment sub-section (page 11), we have edited the text to include the cisplatin schedule and cumulative dose in our population. 

 (page 11, line 221) “The majority of patients undergoing chemotherapy received mostly cisplatin-based chemotherapy (n=42/49; 85.7%), administered 3-weekly during radiotherapy for 3 cycles at 100mg/m2 unless contra-indicated due to inadequate renal function, in which case they received paclitaxel, or carboplatin, or cetuximab instead. The mean cumulative dose of cisplatin administered was 255mg/m2.”

(iii) What is the compliance of RT in the cohort? These treatment-related factors may also have a higher impact on survival. The author may consider to incorporate it into the analysis.

 Compliance to radiotherapy was high with 98% completing RT. We have inserted a sentence regarding this in the Results section (treatment sub-section) on page 13. 

 (page 13, line 251) “Compliance to radiotherapy was high overall, with almost all patients (197/201, 98%) completing RT.” 

(iv) Some studies showed that body weight or BMI may also be a prognostic factor for cancer therapy. Is it possible for the study to retrieve the data from medical record?

 Unfortunately, we do not have the data regarding the patients’ body weight and BMI and hence, we were not able to evaluate the relationship of body weight/ BMI and treatment outcomes. However, we would bear this factor in mind and collect this information prospectively for future studies. 

3. Reviewer #2: 

(i) Too many tables. 

 (a) We have condensed/ combined some of the fields in the Results section, Table 1 (page 10, line 212): 

 - N2 a,b,c, categories are combined under “N2”

 - aCCI score grouped into (0-3) and (4-12)

 (b) We have condensed the univariable and multivariable results for OS, PFS and LFS into 1 table for each outcome 

 • Page 15, line 302: previous table 2A, 2B is now table 3

 • Page 18, line 333: previous table 3A and 3B is now table 4

 • Page 22, line 386: previous table 5A and 5B is now table 5. 

 (c) In the Results section, Organ preservation for laryngeal cancer sub-section (page 20), we have removed table 4 (patient demographics and clinical characteristics of organ preservation/ non-surgical cohort) and instead have highlighted the relevant findings in the paragraph before it.

 (page 20, line 351) “One hundred and seventy out of 215 patients (79.0%) underwent first-line RT or CRT. Compared with patients treated with RT alone, patients treated with CRT were on average younger (age <65 years, 67.4% vs 37% respectively, p<0.001), had lower aCCI score (aCCI ≤3, 86% vs 59.1%, p=0.001 ) and had higher disease stage (stage 3/4, 100% vs 20.5%); T3/4, 69.8% vs 15.7% and N2/3, 41.8% vs 3.9%, p<0.001).”

 (d) In the Results section, Treatment effect of CRT and RT (page 24) we have removed table 6 and incorporated the significant findings in the preceding paragraph. 

 (page 24, line 406) “After accounting for significant prognostic factors, there was a significant treatment effect favoring CRT over RT in terms of OS, PFS and LFS (RT vs CRT, OS HR 1.96 (95% CI: 1.03-3.75; p=0.04); PFS HR 2.33 (95% CI: 1.28 to 4.23; p=0.005); LFS HR 2.14 (95% CI: 1.17 to 3.93; p=0.01).”

 (e) However, due to need to describe the treatment modalities better according to the stage and aCCI, an additional table was included in the Results section, treatment sub-section on page 13 (Table 2: Association of aCCI score with treatment in stage 3/4 patients).

(ii) Treatment should be described according to the TNM, Stage and aCCI as there is 60% T1/T2 and 70% N0. Already known data

 a) We have inserted an additional paragraph under the Results section, treatment sub-section, (page 13-14) describing the breakdown of treatment according to stage of disease. We have also sought to explain why a proportion of stage 3/4 patients were treated with single modality treatment. 

 (page 13, line 260) “Almost all of the patients with stage 1/2 disease received single modality treatment with primary RT alone (101/106; 95.3%), of the remainder, 4 patients (3.8%) received surgery alone and 1 received surgery followed by RT. Amongst 109 stage 3/4 patients, two-thirds (73 patients, 67%) received combined modality treatment: in particular, 43 (39.4%) received primary CRT, 24 (22.0%) underwent surgery followed by adjuvant RT whilst 6 patients (5.5%) had surgery followed by adjuvant CRT. The remaining one-third of patients (n=36, 33%) received single modality treatment (surgery or RT): 26 (23.9%) received radiation alone: of these, 12 (46%) were elderly (>70 years old), 9 (35%) had multiple comorbidities; these patients were deemed unsuitable for CRT, whilst 4 (15%) declined chemotherapy and the remaining patient was treated in 1997, when CRT was not yet standard of care. 10 patients underwent surgery alone: 3 declined adjuvant RT, 3 had received previous RT to that region for a separate unrelated tumour (nasopharyngeal carcinoma), 2 patients defaulted RT, 1 patient had a synchronous carcinoma of the colon requiring treatment, and 1 had post-operative wound infection which precluded RT.”

 b) In addition, under the same section, page 14, we have inserted a paragraph describing the association of aCCI score with treatment in stage 3/4 patients and inserted a new table (table 2) reporting the results of this analysis. 

 (page 14, line 275)“Amongst stage 3/4 patients, the treatment modality (single vs combination) was associated with the aCCI score at diagnosis (p<0.001) and a test for trend showed evidence of a linear trend, with the proportion of patients receiving single modality treatment increasing as the aCCI score increased (p<0.001) (Table 2).”

We have also inserted an additional sentence in the Methodology section, “Statistical analysis” sub-section (page 9) detailing the methods used for the above analysis: 

 (page 9, line 184) “The chi-squared test for trend was used to test if there was a linear trend relating aCCI score to treatment decision (single/combination modality).”

 c) Whilst some of the data presented is already known, however, there are limited studies in Asian populations and also limited data regarding the association of aCCI and outcomes such as PFS and LFS in laryngeal cancer. Furthermore, the association of aCCI and treatment modality presented here has not been widely reported, and to some extent represents ‘real-world’ experience and points towards an area where more efforts could be directed. 

4. Reviewer #3: 

(i) The major concern of this study is less novelty, since comorbidities and nodal status have both been correlated with treatment outcome in many cancers including HNSCC.

 As mentioned above (3 (ii)(c)), whilst some of the data presented is already known, however there are limited studies of comorbidity in Asian populations particularly in head and neck cancer, and also limited data regarding its association with outcomes such as progression-free survival and laryngectomy-free survival. Furthermore, the association of co-morbidity and treatment modality received has not been widely reported, and to some extent represents ‘real-world’ experience and points towards an area where more efforts could be directed. 

(ii) The treatment strategy is heterogenous and therefore make each sub-group with small number. Authors could conduct an analysis with a more specific population, e.g., locally advanced LSCC after primary chemoradiation with IMRT and triweekly cisplatin.

 We agree with the reviewer that the study treatment is heterogenous and leads to many small treatment sub-groups which limits the study findings and interpretation. Unfortunately, within this particular study cohort, further analysis of treatment sub-groups such as locally advanced LSCC with primary CRT will not be feasible due to small patient numbers. However, based on our study findings so far, we are hoping to proceed with further evaluation in a larger and homogenous cohort of patients 

(iii) The enrolment duration is across almost 20 years, while there may be some prognosis shift either due to treatment or even general care improvement.

 We agree with the reviewer’s comments. As mentioned in (1), we have included a paragraph in the Discussion section (page 28, line 497) summarising the changes in management of laryngeal cancer in the past 20 years. Ideally, we would like to analyse the study findings according to time periods, however this is not possible due to small patient numbers when patients are divided into the different time periods.

(iv) More than 2/3 of patients only accepted RT alone. As the authors discussed, they also need to explain why these patients have more comorbidities, poor performance status, or advanced age.

 We have inserted an additional paragraph under the Results section (treatment sub-section, page 13) describing the breakdown of treatment according to stage of disease. The majority of patients who received RT alone had stage 1/2 disease (101/127, 79.5%). For the remaining patients with stage 3/4 disease (26/127, 20.5%) we have explained the reasons they were treated with RT alone (page 13). 

 (page 13, line 266) “26 (23.9%) received radiation alone: of these, 12 (46%) were elderly (>70 years old), 9 (35%) had multiple comorbidities; these patients were deemed unsuitable for CRT, whilst 4 (15%) declined chemotherapy and the remaining patient was treated in 1997, when CRT was not yet standard of care.”

We hope that we have addressed all the issues raised, and would be happy to clarify further on any other issues. Thank you for kind review and for consideration of our manuscript for publication in your journal.

---

## [Editor Report · Decision Letter 1]

21 Oct 2019

Association of clinical factors with survival outcomes in laryngeal squamous cell carcinoma (LSCC)

PONE-D-19-20376R1

Dear Dr. Ang, 

We are pleased to inform you that your manuscript has been judged scientifically suitable for publication and will be formally accepted for publication once it complies with all outstanding technical requirements.

With kind regards,

Jason Chia-Hsun Hsieh, M.D. Ph.D

Academic Editor

PLOS ONE

Additional Editor Comments (optional):

All the questions have been answered adequately.
---

## [Editor Report · Acceptance letter]

7 Nov 2019

PONE-D-19-20376R1 

Association of clinical factors with survival outcomes in laryngeal squamous cell carcinoma (LSCC) 

Dear Dr. Ang:

I am pleased to inform you that your manuscript has been deemed suitable for publication in PLOS ONE. Congratulations! Your manuscript is now with our production department. 

With kind regards,

on behalf of

Dr. Jason Chia-Hsun Hsieh 

Academic Editor

PLOS ONE